# The Antidiabetic Agent Metformin Inhibits IL-23 Production in Murine Bone-Marrow-Derived Dendritic Cells

**DOI:** 10.3390/jcm10235610

**Published:** 2021-11-29

**Authors:** Tomoyo Matsuda-Taniguchi, Masaki Takemura, Takeshi Nakahara, Akiko Hashimoto-Hachiya, Ayako Takai-Yumine, Masutaka Furue, Gaku Tsuji

**Affiliations:** 1Department of Dermatology, Graduate School of Medical Sciences, Kyushu University, Fukuoka 812-8582, Japan; taniguchi.tomoyo.735@s.kyushu-u.ac.jp (T.M.-T.); take0917@dermatol.med.kyushu-u.ac.jp (M.T.); nakahara.takeshi.930@m.kyushu-u.ac.jp (T.N.); ahachi@dermatol.med.kyushu-u.ac.jp (A.H.-H.); a-takai@med.kyushu-u.ac.jp (A.T.-Y.); furue@dermatol.med.kyushu-u.ac.jp (M.F.); 2Research and Clinical Center for Yusho and Dioxin, Kyushu University Hospital, Fukuoka 812-8582, Japan

**Keywords:** BMDCs, IL-23, IL-36γ, psoriasis, metformin

## Abstract

Psoriasis is a chronic inflammatory skin disease, and its immune mechanism has been profoundly elucidated. Biologics targeting interleukin (IL)-23 have prevented the development of psoriasis. As major sources of IL-23, dendritic cells (DCs) play a pivotal role in psoriasis; however, the regulatory mechanism of IL-23 in DCs remains unclear. IL-36γ was reported to reflect the disease activity of psoriasis. Therefore, we hypothesized that IL-36γ may affect IL-23 production in DCs. To reveal the mechanism by which IL-36γ controls IL-23 production in DCs, we analyzed murine bone marrow-derived DCs (BMDCs) stimulated with IL-36γ. IL-36γ stimulation upregulated the mRNA and protein expression of Nfkbiz in BMDCs. Nfkbiz knockdown using siRNA transfection partially inhibited the upregulation of IL-23 mRNA expression induced by IL-36γ stimulation. Since NF-κB signaling regulates Nfkbiz expression and the anti-diabetic agent metformin reportedly modulates NF-κB signaling, we examined the effect of metformin treatment on IL-36γ-induced IL-23 production. Metformin treatment impaired the phosphorylation of NF-κB induced by IL-36γ stimulation with the subsequent downregulation of Nfkbiz, resulting in the inhibition of IL-23 production in BMDCs. These data provided evidence that metformin treatment can inhibit IL-36γ-mediated IL-23 production in BMDCs, which might contribute to the prevention of psoriasis.

## 1. Introduction

Psoriasis is an immune-mediated inflammatory skin disease affecting 2–4% of the global population [1]. The skin lesions in psoriasis manifest desquamative erythema, which profoundly impairs patients’ quality of the life [2]. The pathology of psoriasis is characterized by epidermal hyperproliferation, the intraepidermal accumulation of neutrophils, and the infiltration of dermal inflammatory cells such as T-cells, macrophages, and dendritic cells (DCs) [3]. Among these immune cells, DC counts are increased significantly in psoriatic lesions [4]. Furthermore, autoantigens from keratinocytes activate plasmacytoid DCs (pDCs) in the dermis. pDCs produce type I interferon and tumor necrosis factor-α, which activates classical DCs (cDCs), resulting in interleukin (IL)-23 secretion. IL-23 is mostly produced by cDCs, which correspond to CD1c+ DCs in humans. IL-23 by cDCs promotes Th17 differentiation in mice and humans [5]. In a murine model of psoriasis induced by topical imiquimod (IMQ), DCs were identified as the major source of IMQ-induced IL-23, which is critical for the development of psoriatic skin lesions [6,7]. Several reports have shown that IL-23 derived from DCs is involved in the pathogenesis of psoriasis [8,9,10]. Therefore, IL-23-producing DCs play a central role in the pathogenesis of psoriasis, which supports clinical evidence that the administration of monoclonal antibodies against IL-23 such as guselkmab, rizankizumab, and tildrakizumab can facilitate the achievement of a Psoriasis Area and Severity Index 90 response at week 16 (67–75%) [11,12,13]. Whereas IL-23 is a key regulator of IL-17 production, the mechanism of IL-23 production by DCs in psoriasis remains unclear. Recently, serum levels of IL-36γ, a member of the IL-36 family, were identified as a disease activity marker of psoriasis [14]. Meanwhile, IL-36γ is highly expressed in the epidermis in psoriatic lesions [15]. As IL-36γ derived from keratinocytes potentially activates DCs [16], we hypothesized that IL-36γ is involved in IL-23 production in DCs during the pathogenesis of psoriasis. To test this hypothesis, we analyzed murine bone marrow-derived DCs (BMDCs) stimulated with IL-36γ.

Furthermore, we examined whether metformin, an antidiabetic agent, modulates IL-36γ signaling in BMDCs. Metformin is mostly used to treat type 2 diabetes (T2DM), and a high prevalence of T2DM in patients with severe psoriasis has been identified [17]. In clinical studies of patients with psoriasis, long-term treatment with metformin has been shown to reduce the risk of psoriasis [18]. In addition, metformin administration has been shown to improve the severity of psoriasis [19,20]. These clinical results support the likelihood that metformin treatment is effective against both psoriasis and T2DM; however, the molecular mechanism remains unknown.

## 2. Materials and Methods

### 2.1. Reagents and Antibodies

Anti-murine NF-κB p65 monoclonal rabbit antibody (Abcam, Cambridge, UK), anti-murine NF-κB p65 (phospho Ser536) polyclonal rabbit antibody (Abcam), anti-murine IκBζ (protein corded by NFKBIZ gene) polyclonal rabbit antibody, and anti-murine β-actin monoclonal mouse antibody (Cell Signaling Technology, Danvers, MA, USA) were used for Western blotting. Dimethyl Sulfoxide (DMSO) was purchased from Nacalai Tesque, Inc. (Kyoto, Japan). Metformin hydrochloride and BAY 11-7082 were obtained from Tokyo Chemical Industry Co., Ltd. (Tokyo, Japan). Murine recombinant IL-36γ was obtained from R&D Systems (Minneapolis, MN, USA).

### 2.2. Generation of BMDCs and Cell Culture

C57BL/6N female mice were housed in a clean facility until 6 weeks of age by CLEA Japan, Inc. (Fujinomiya, Japan). The animal experiments were conducted in accordance with a protocol reviewed and approved by the animal facility center of Kyushu University (A21-283-0, 2021–2023). Bone marrow cells freshly isolated from the femoral and tibial bones of mice were cultured in RPMI 1640 medium (Merck KGaA, Darmstadt, Germany) containing 1 mmol/L sodium pyruvate (Thermo Fisher Scientific, Waltham, MA, USA), 10 mmol/L 4-(2-hydroxyethyl)-1-piperazineethanesulfonic acid (Thermo Fisher Scientific), 1% Minimum Essential Medium Non-Essential Amino Acids (Thermo Fisher Scientific), 10% FBS (Capricorn Scientific GmbH, Ebsdorfergrund, Germany), 50 nmol/L β-mercaptoethanol (Nacalai Tesque), and antibiotic–antimycotic 100× (100 U/mL penicillin, 100 mg/mL streptomycin, and 0.25 μg/mL amphotericin B; Thermo Fisher Scientific) containing GM-CSF (10 ng/mL) (PeproTech, Cranbury, NJ, USA). On day 3, half of the culture medium and GM-CSF were added. On day 5, non-adherent cells were subcultured, and GM-CSF was added. On day 7, half of the culture medium and GM-CSF were added. On day 9, non-adherent cells were harvested. These cells were purified immunomagnetically via three rounds of positive selection with CD11c (N418) MicroBeads (Miltenyi Biotec, Bergisch Gladbach, Germany). Purified BMDCs were cultured with/without stimulants such as IL-36γ, metformin, and BAY 11-7082 for the indicated times. Culture supernatant was collected after 24 h and analyzed by ELISA. Cells were also collected for quantitative reverse transcription (qRT)-PCR or Western blotting.

### 2.3. Transfection of Small Interfering RNAs (siRNAs) against Nfkbiz

siRNAs against Nfkbiz and non-targeting siRNA (control siRNA) were obtained from Thermo Fisher Scientific. Cells were incubated in culture medium with a mixture containing 300 nM siRNA for transfection using program DK-100 following the Amaxa^®^ 4D-Nucleofector^®^ Protocol for Immature Mouse Dendritic Cells For 4D-Nucleofector^®^ X Unit (Lonza Group AG, Basel, Switzerland).

### 2.4. qRT-PCR

Total RNA was extracted using an RNeasy^®^ Mini kit (Qiagen, Venlo, The Netherlands). Reverse transcription was performed using a PrimeScript™ RT reagent kit (Takara Bio, Shiga, Japan). qRT-PCR was conducted on a CFX Connect™ Real-time System (Bio-Rad, Hercules, CA, USA). Gene expression levels of IL-23 and Nfkbiz were determined by qRT-PCR using TaqMan Fast Advanced Master Mix (Thermo Fisher Scientific). Amplification was initiated at 95 °C for 20 s as the first step, followed by 40 cycles of qRT-PCR at 95 °C for 3 s and at 60 °C for 30 s as the second step. mRNA expression was measured in triplicate with normalization by the housekeeping gene Ywhaz, and expression was indicated as the fold change relative to the control group. Primer sequences are listed in Appendix A.

### 2.5. Western Blotting

Cells were incubated for 5 min in cOmplete™ Lysis-M (Roche Diagnostics, Basel, Switzerland). The protein concentration in the lysate was measured using a BCA Protein Assay Kit (Thermo Fisher Scientific). Equal amounts of protein (15 μg) were dissolved in Bolt LDS sample buffer (Thermo Fisher Scientific) and a 10% sample reducing agent (Thermo Fisher Scientific). The lysates were boiled at 70 °C for 10 min and then to electrophoresis in NuPAGE 4–12% Bis-Tris gels (Thermo Fisher Scientific) at 200 V for 25 min. The proteins were then transferred onto polyvinylidene difluoride membranes (Thermo Fisher Scientific), which were blocked with WesternBreeze Blocker/Diluent (Thermo Fisher Scientific). The membranes were then probed with anti-murine NF-κB p65 monoclonal rabbit antibody, anti-murine NF-κB p65 (phospho Ser536) polyclonal rabbit antibody, and anti-murine IκBζ polyclonal rabbit antibody (all from Cell Signaling Technology) overnight at 4 °C. Horseradish peroxidase-conjugated anti-rabbit IgG antibodies (Cell Signaling Technology) served as secondary antibodies. Protein bands were visualized with Chemi-Lumi One Super (Nacalai Tesque) using the ChemiDoc touch imaging system (Bio-Rad). The membranes were then re-blotted with Restore™ PLUS Western Blot Stripping Buffer (Thermo Fisher Scientific) and anti-murine β-actin mouse antibody 30 min at room temperature. Horseradish peroxidase-conjugated anti-mouse IgG antibodies served as secondary antibodies. Protein bands were visualized with SuperSignal™ West Pico PLUS Chemiluminescent Substrate (Thermo Fisher Scientific) using the ChemiDoc touch imaging system (Bio-Rad). Densitometric analysis of the bands was performed using ImageJ software. ImageJ is a public domain, Java-based image processing program developed at the National Institutes of Health (Bethesda, MD, USA). Experiments were repeated three times in separate experiments.

### 2.6. ELISA

A murine IL-23 ELISA Kit (R&D Systems) was used for ELISA in accordance with the manufacturer’s protocol. Optical density was measured using a DTX 800 Multimode Detector (Beckman Coulter, Brea, CA, USA).

### 2.7. Statistical Analysis

Statistical analysis was performed with GraphPad Prism 5.0 (GraphPad Software, Inc., La Jolla, CA, USA). An unpaired Student’s *t*-test was used to analyze the results, and a *p*-value of less than 0.05 was considered statistically significant.

## 3. Results

### 3.1. IL-36γ Stimulation Upregulated IL-23 and Nfkbiz Expression in BMDCs

To investigate the mechanism by which IL-36γ regulates IL-23 expression in DCs, we analyzed murine BMDCs stimulated with IL-36γ. qRT-PCR analysis revealed that IL-36γ stimulation (100 ng/mL) for 1, 2, 4, or 6 h upregulated IL-23 mRNA expression with expression peaking at 1 h (Figure 1A). Additionally, IL-36γ stimulation (1, 10, 50, or 100 ng/mL) for 1 h upregulated IL-23 mRNA expression in a concentration-dependent manner (Figure 1B). Furthermore, ELISA of the culture medium of BMDCs stimulated with IL-36γ (1, 10, 50, or 100 ng/mL) for 24 h revealed IL-23 production in a concentration-dependent manner (Figure 1C). We measured IL-23 production by ELISA following stimulation for 1 or 6 h; however, IL-23 was undetectable (data not shown). We believe that 24 h are required for IL-23 secretion to proceed after IL-23 mRNA expression is increased. As NFKBIZ is reported to be a key transcriptional regulator of IL-36-related gene expression in human psoriatic keratinocytes [21,22], we evaluated Nfkbiz expression in addition to IL-23 expression in murine BMDCs. qRT-PCR analysis illustrated that IL-36γ stimulation upregulated Nfkbiz mRNA expression (Figure 1D), which was in a concentration-dependent manner (Figure 1E). Western blotting analysis confirmed that IL-36γ stimulation (100 ng/mL) for 1, 2, 4, or 6 h upregulated IκBζ (protein corded by Nfkbiz gene) protein expression (Figure 1F).

### 3.2. IL-36γ Stimulation Upregulated IL-23 via Nfkbiz in BMDCs

Next, we examined whether Nfkbiz is involved in IL-23 upregulation induced by IL-36γ in murine BMDCs. We transfected BMDCs with either scrambled siRNA (si-control) or siRNA targeting Nfkbiz (si-Nfkbiz) and then stimulated the cells with IL-36γ (10 ng/mL) for 1 h. Although the transfection of si-Nfkbiz alone did not alter mRNA and IκBζ protein expression in BMDCs, it successfully downregulated Nfkbiz mRNA (Figure 2A) and protein expression (Figure 2B) in BMDCs stimulated with IL-36γ. This finding may be related to the partial depletion of the target gene because siRNA transfection is difficult in DCs [23]. Furthermore, we observed that depletion of Nfkbiz via siRNA transfection partially canceled IL-36γ stimulation-induced IL-23 mRNA upregulation (Figure 2C). Although we attempted to measure IL-23 production in the culture supernatant of siRNA-transfected BMDCs using ELISA, we could not detect IL-23 production, which may be attributable to cell damage caused by the siRNA transfection procedure. These results suggest that IκBζ is likely an integral part of the IL-36γ-induced IL-23 upregulation in murine BMDCs.

### 3.3. IL-36γ Upregulates Nfkbiz and IL-23 via the Activation of NF-κB Signaling

Next, we examined the mechanism by which IL-36γ upregulates Nfkbiz expression in murine BMDCs. Considering that IL-36 binding to the IL-36 receptor complex leads to the recruitment of MyD88 and activation of NF-κB signaling [16] and that Nfkbiz expression is regulated by phosphorylation of p65, a component of the NF-κB heterodimer [24], we hypothesized that IL-36γ modulated Nfkbiz expression via NF-κB signaling in murine BMDCs. We analyzed p65 phosphorylation in murine BMDCs stimulated with IL-36γ (100 ng/mL) for 10, 20, 30, 40, or 60 min using Western blotting. We confirmed that p65 phosphorylation was induced after 10 min of IL-36γ stimulation (Figure 3A). We further examined whether BAY 11-7082, an inhibitor of p65 phosphorylation, affects the upregulation of Nfkbiz induced by IL-36γ stimulation. We stimulated murine BMDCs with IL-36γ (100 ng/mL) for 1 h in the absence or presence of BAY 11-7082 (10, 50, or 100 μM) and measured Nfkbiz mRNA and IκBζ protein expression by qRT-PCR (Figure 3B) and Western blotting (Figure 3C), respectively. BAY 11-7082 treatment inhibited Nfkbiz upregulation in a concentration-dependent manner (Figure 3B,C). Moreover, we examined whether BAY 11-7082 treatment inhibits the upregulation of IL-23 induced by IL-36γ. We measured IL-23 production in the culture supernatant of BMDCs stimulated with IL-36γ (100 ng/mL) for 24 h in the absence or presence of BAY 11-7082 (10, 50, or 100 μM) using ELISA. BAY 11-7082 treatment also inhibited the upregulation of IL-23 induced by IL-36γ stimulation in a concentration-dependent manner (Figure 3D).

### 3.4. Metformin Treatment Inhibited the Upregulation of Nfkbiz and IL-23 Induced by IL-36 Stimulation by Impairing NF-κB Signaling

It has been reported that metformin controls NF-κB signaling [25]. Moreover, clinical studies of patients with psoriasis and T2DM have suggested that metformin administration may attenuate the disease activity of psoriasis [19,20]. Therefore, we hypothesized that metformin treatment affects the IL-36γ-induced upregulation of Nfkbiz and IL-23 by modulating NF-κB signaling in murine BMDCs. To test this, we analyzed p65 phosphorylation in BMDCs stimulated with IL-36γ (100 ng/mL) for 10, 20, 30, 40, and 60 min in the absence or presence of metformin (5 mM) using Western blotting. Metformin treatment inhibited p65 phosphorylation induced by IL-36γ (Figure 4A). In addition, we evaluated mRNA and protein expression in murine BMDCs stimulated with IL-36γ (100 ng/mL) for 1 h in the absence or presence of metformin (0.5, 1, or 5 mM). Metformin treatment inhibited the IL-36γ-induced upregulation of Nfkbiz mRNA and IκBζ protein expression in a concentration-dependent manner in BMDCs (Figure 4B,C). Subsequently, metformin treatment was revealed to inhibit the upregulation of IL-23 mRNA expression in BMDCs stimulated with IL-36γ for 1 h in a concentration-dependent manner (Figure 4D). In addition, metformin treatment downregulated IL-23 production in the culture supernatant of BMDCs stimulated with IL-36γ for 24 h in a concentration-dependent manner (Figure 4E).

## 4. Discussion

IL-36γ is a cytokine associated with the disease activity of psoriasis. Therefore, it is of great interest to identify a strategy that inhibits the responses of downstream inflammatory cytokines. In this study, we obtained evidence that IL-36γ induces Nfkbiz upregulation, which subsequently leads to IL-23 upregulation, in murine BMDCs. Furthermore, we revealed that metformin treatment inhibited IL-23 upregulation via the impairment of Nfkbiz upregulation. To our knowledge, this is the first study providing evidence that metformin can modulate IL-36γ-mediated IL-23 production in DCs, thereby contributing to the prevention of psoriasis. However, the systemic expression of IL-23 following the administration of IL-23 minicircle DNA [26] or transgenic expression of IL-23 derived from keratinocytes [27] can reportedly promote the development of psoriasis. Therefore, the production of IL-23 by cells other than DCs has also been reported to contribute to the pathogenesis of psoriasis, which is a limitation of this study.

We confirmed that IL-36γ (10 or 100 ng/mL) stimulation efficiently induces IL-23 mRNA upregulation in murine BMDCs, which is consistent with previous findings [28,29] Furthermore, we revealed that IL-36γ stimulation increases IL-23 protein expression using ELISA. Although the concentration of 100 ng/mL is rather high, we considered it reasonable in this experiment system because IL-36γ is activated by neutrophil-derived proteases such as proteinase-3 [30].

IκBζ, a transcriptional regulator of selective NF-κB target genes, has been identified as a crucial mediator of IL-36-driven psoriasis-related gene expression in keratinocytes [21]; however, it is unclear whether IκBζ is involved in the IL-36γ-mediated expression of genes including IL-23 in DCs. Based on the result that depletion of Nfkbiz mRNA expression partially downregulated IL-23 mRNA expression induced by IL-36 stimulation, IκBζ is likely an integral part of IL-36-induced IL-23 upregulation in murine BMDCs. As we utilized siRNA transfection to deplete Nfkbiz mRNA expression, the low depletion efficiency may have resulted in weak repression of IL-23 mRNA expression.

Although several studies suggested that the molecular machinery underlying the regulation of IκBζ could be cell type-specific, it has been reported that the induction of IκBζ by IL-36γ is mediated by MyD88, NF-κB, and STAT3 in human keratinocytes [13,14]. We found that inhibition of NF-κB activity by BAY 11-7082 treatment significantly downregulated IκBζ induction by IL-36γ in murine BMDCs, suggesting that NF-κB activation may be critical in IL-36γ-mediated IκBζ induction in DCs. Furthermore, inhibition of NF-κB activity by metformin treatment had the same effect as BAY 11-7082 treatment. These data support the potential benefits of metformin treatment in patients with type 2 diabetes mellitus (T2DM) and psoriasis. As several studies revealed an association between T2DM and psoriasis [31,32] and suggested a severity-dependent relationship between psoriasis and T2DM [17], the management of T2DM is considered extremely important in the treatment of psoriasis. Metformin treatment has been reported to inhibit several signaling pathways including the mammalian target of rapamycin [33] and mitogen-activated protein kinase signaling [34], in addition to NF-κB signaling. As such, we cannot thus exclude the possibility that other kinases might have been affected by metformin in the experiments. Further studies will be required to clarify this possibility.

We revealed here that metformin exerts anti-inflammatory effects on DCs, at least in part via pathways involving the inhibition of IκBζ production (Figure 4F). Our study presents the novel concept that pharmacological modulation by metformin of IL-36γ-induced IL-23 production via IκBζ inhibition may offer a potential therapeutic approach to psoriasis. It can be hypothesized that oral metformin administration suppresses psoriasis by this mechanism. However, the clinical relevance of IL-23 inhibition by metformin requires further investigation.

## Figures and Tables

**Figure 1 jcm-10-05610-f001:**
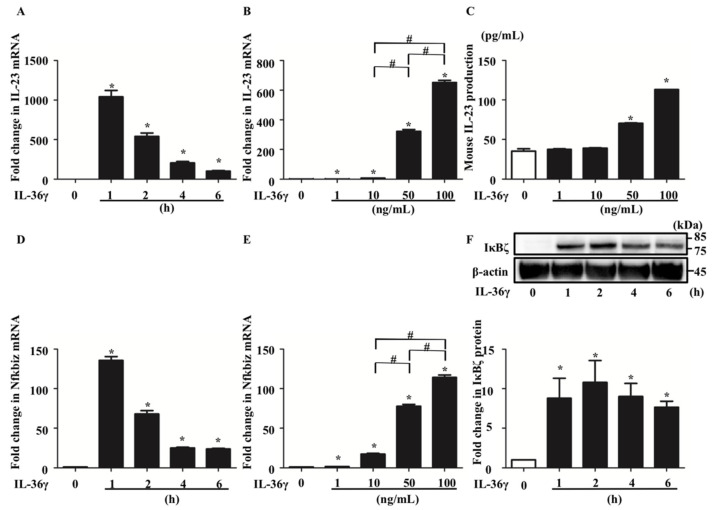
IL-36γ stimulation upregulated IL-23 and Nfkbiz in bone marrow-derived dendtitic cells (BMDCs). (**A**,**D**,**F**) BMDCs were stimulated with IL-36γ (100 ng/mL) for 1, 2, 4, or 6 h. (**A**,**D**) Quantitative reverse transcription (qRT)-PCR. (**F**) Western blotting. IκBζ protein levels are normalized to β-actin protein levels using ImageJ and expressed as fold change. (**B**,**C**,**E**) BMDCs were stimulated with IL-36γ (1, 10, 50, or 100 ng/mL) for 1 h. (**B**,**C**) qRT-PCR. (**E**) BMDCs were stimulated with IL-36γ (100 ng/mL) for 24 h, and IL-23 production in the culture supernatant was measured by ELISA. (**A**–**D**) Data are expressed as the mean ± standard error of the mean (SEM); *n* = 3/group. * Significant differences between the IL-36γ-stimulated groups and control groups (*p* < 0.05). # Significant differences between the IL-36γ-stimulated groups of each dose (*p* < 0.05). mRNA levels normalized for Ywhaz expression were expressed as the fold change compared to that in the control group. (**C**) Data are expressed as the mean ± SEM; *n* = 3/group; * *p* < 0.05. (**F**) Data are representative of experiments repeated three times with similar results.

**Figure 2 jcm-10-05610-f002:**
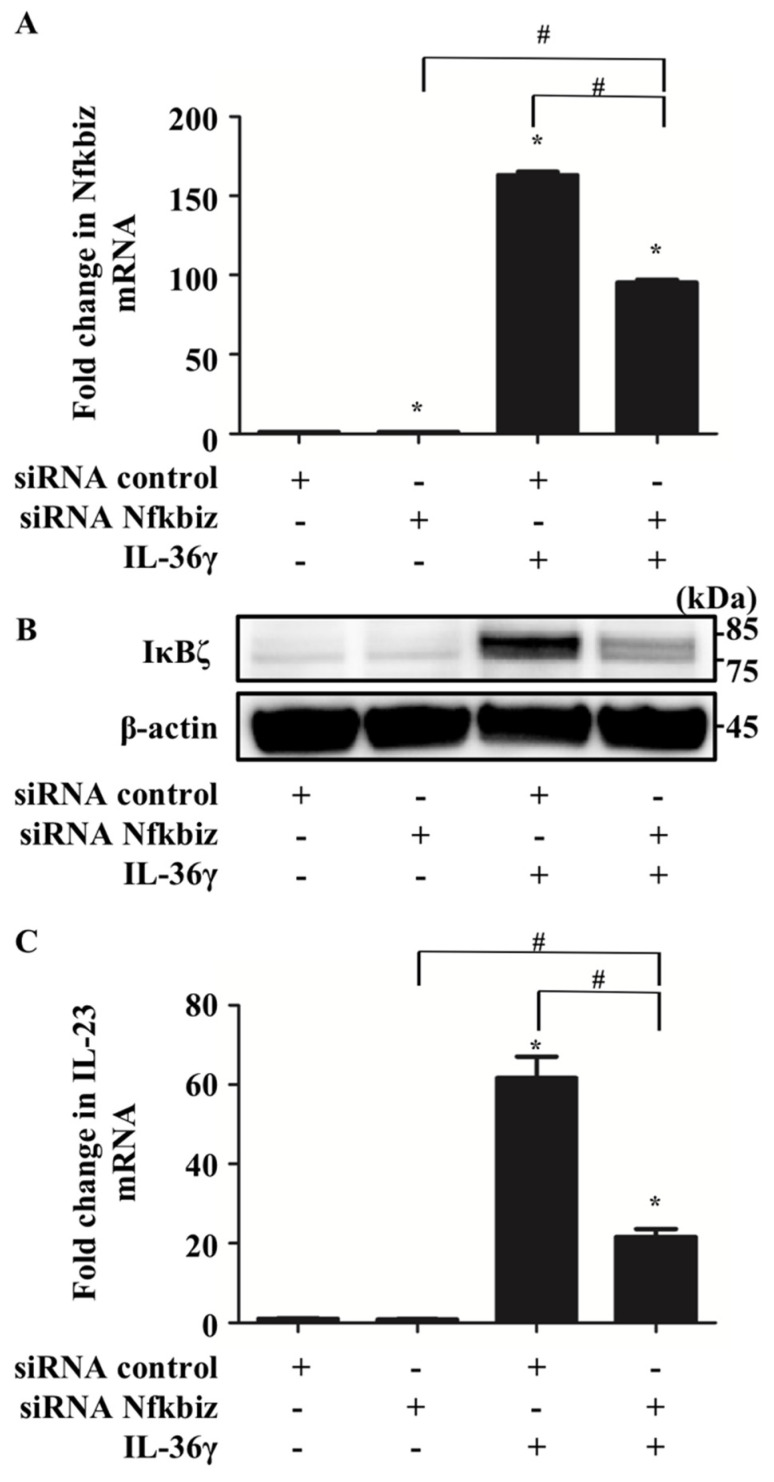
IκBζ is likely an integral part of IL-36γ-induced IL-23 upregulation in BMDCs. Control small interfering RNA (siRNA)- or Nfkbiz siRNA-transfected BMDCs were treated with/without IL-36γ (10 ng/mL) for 1 h and analyzed via quantitative reverse transcription (qRT)-PCR and Western blotting. +/− indicates whether siRNA or IL-36γ is utilized. (**A**) qRT-PCR. (**B**) Western blotting. (**C**) qRT-PCR. (**A**,**C**) Data are expressed as the mean ± standard error of the mean (SEM); *n* = 3/group. * Significant difference versus the control siRNA-transfected group with no IL-36γ stimulation (*p* < 0.05). # Significant difference between the Nfkbiz siRNA-transfected and control siRNA-transfected groups that were stimulated with IL-36γ (*p* < 0.05). mRNA levels normalized to Ywhaz mRNA expression are expressed as the fold change versus that in the control group. (**B**) IκBζ expression was evaluated using anti-murine IκBζ antibody. Data are representative of experiments repeated three times with similar results.

**Figure 3 jcm-10-05610-f003:**
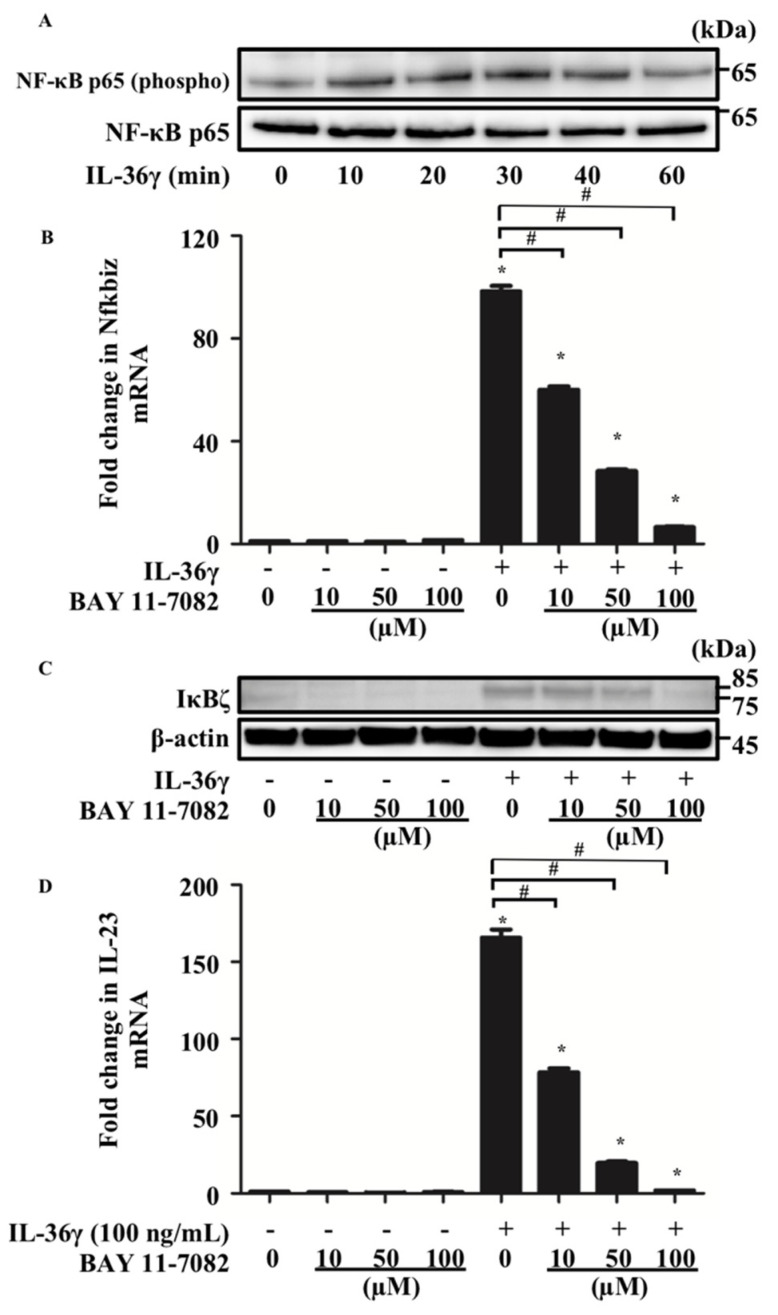
Nfkbiz expression was regulated by p65 phosphorylation in BMDCs. BMDCs were stimulated with IL-36γ (100 ng/mL) for 10, 20, 30, 40, or 60 min (**A**). (**A**) Western blotting. BMDCs were stimulated with/without IL-36γ (100 ng/mL) for 1 h (+/−) in the absence or presence of BAY 11-7082 (10, 50, or 100 μM) (**B**–**D**). (**B**,**D**) Quantitative reverse transcription-PCR. (**C**) Western blotting. (**B**,**D**) Data are expressed as the mean ± standard error of the mean; *n* = 3/group. * Significant difference between the IL-36γ-stimulated and control groups (*p* < 0.05). # Significant difference between the BAY 11-7082-treated and untreated groups that were stimulated with IL-36γ (*p* < 0.05). mRNA levels normalized to Ywhaz mRNA expression were expressed as the fold change versus that in the control group. (**A**,**C**) Data are representative of experiments repeated three times with similar results. +/− indicates whether IL-36γ is utilized.

**Figure 4 jcm-10-05610-f004:**
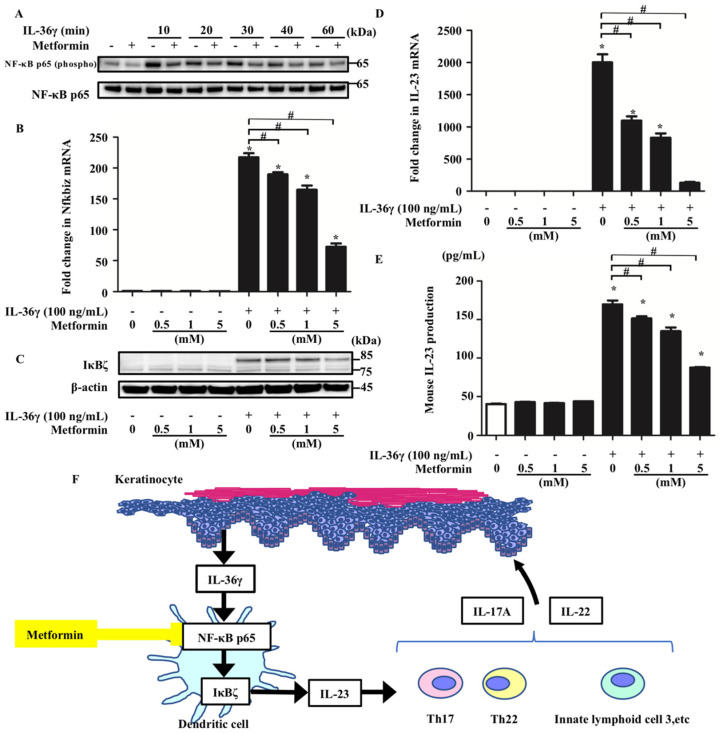
Metformin treatment inhibited IL-36γ-induced upregulation of Nfkbiz and IL-23 by modulating NF-κB signaling in BMDCs. BMDCs stimulated with IL-36γ (100 ng/mL) for 10, 20, 30, 40, or 60 min in the absence or presence of metformin (5 mM) (**A**). (**A**) Western blotting. BMDCs were stimulated with/without IL-36γ (100 ng/mL) for 1 h (+/−) in the absence or presence of metformin (0.5, 1, or 5 mM). (**B**,**D**,**E**). (**B**,**D**) quantitative reverse transcription-PCR. (**E**) ELISA. (**F**) Graphical abstract. Data are expressed as the mean ± standard error of the mean; *n* = 3/group. * Significant difference between the IL-36γ-stimulated and control groups (*p* < 0.05). # Significant differences between the metformin-treated and control groups that were stimulated with IL-36γ (*p* < 0.05). mRNA levels normalized for Ywhaz expression were expressed as fold changes versus that in the control group. (**A**,**C**) Data are representative of experiments repeated three times with similar results. +/− indicates whether IL-36γ or metformin is utilized.

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
