# Peer review of "The Antidiabetic Agent Metformin Inhibits IL-23 Production in Murine Bone-Marrow-Derived Dendritic Cells"

_jcm, 2021, doi:10.3390/jcm10235610_

Round 1
Reviewer 1 Report
I believe now the paper might be published in JCM
Author Response
Thank you very much for your kind reply.
Reviewer 2 Report
The authors of the submitted manuscript investigate the mechanism by which IL-36 gamma regulates IL-23 production in DCs. They suggest that IL-36 gamma stimulation upregulates Nfkbiz expression in BMDCs. The authors further showed that metformin, the antidiabetic agent, impairs the phosphorylation of NF-κB induced by IL-36 gamma stimulation with the subsequent downregulation of Nfkbiz, resulting in the inhibition of IL-23 production in BMDCs. They found a very interesting molecular for potential treatment for psoriasis.
Specific comments:
1, As we know, not only DC produced IL-23 contributes to pathogenesis of psoriasis. It has been reported that systemic expression of IL-23 resulting from the administration of IL-23 minicircle DNA (DOI: 10.1038/nm.2817; DOI: 10.1016/j.cmet.2021.04.015) and transgenic expression of IL-23 in keratinocytes (DOI: 10.1038/s41598-020-65269-6) can also cause psoriasis. Kindly acknowledge them in the revised manuscript.
2, In the page 1, line 24-25, the authors conclusion is too definitive. It should be read “… may/ might contribute to prevention of psoriasis. ”
3, Figure 1A shows IL-36 gamma stimulation upregulated IL-23 expression in different time points and peak is about 1 h. If this is true, in Figure 1C, why they perform ELISA after 24h stimulation?
4, In Figure 1F, authors need to quantification of the western blots results.
5, Figure 2C, they observed that depletion of Nfkbiz via siRNA transfection partially canceled IL-36 gamma stimulation-induced IL-23 mRNA upregulation. They have methods to detect IL-23 in protein level. The authors should address this by ELISA.
6, Figure 4D, BMDCs stimulated with IL-36 gamma (100ng/ml) for 24 h significantly increased IL-23 mRMA expression without metformin (0 mM) (2000 fold.). However, in Figure 1A, DCs stimulated with IL-36 gamma (100ng/ml) for 6 h increased IL-23 mRMA expression with almost 100 fold. How can authors explain this discrepancy?
7, Figure 4F, first, in addition to IL-17, IL-22, downstream cytokine of IL-23, may also contribute to psoriasis. More than this, IL-23 can induce T cell-independent immune response contributing to skin inflammation (DOI: 10.1172/jci37378). The authors should amend this in graph abstract.
Author Response
Please see the attachment.

This manuscript is a resubmission of an earlier submission. The following is a list of the peer review reports and author responses from that submission.
Round 1
Reviewer 1 Report
The authors' of the reviewed manuscript aimed to evaluate the role of Il36 in Il23 secretion by dendritic cells. While some interesting notion emerged a couple of issues remained to be solved.
- In the introduction, no explicit statement about the role of metformin is recalled, though the title seems to majorly promote it. Please revise this.
- No info's regarding mouse sex, breeding, randomization is provided.
- No recall to ARVO statement for animal use Is reported.
- No power calculation analysis has been performed, which may in turn affect the reliability of the proposed results.
- The evaluation of the concentration of different mediators is not consistent throughout the paper (sometimes the authors used elisa, sometimes WB). RT-PCR for the same mediators has been used seldom. Why did the authors choose different approaches?
- Conclusions might be better organised.
Reviewer 2 Report
This is an interesting article which describes "metformin treatment impaired the phosphorylation of NF-κB induced by IL-36γ stimulation 23
with the subsequent downregulation of Nfkbiz, resulting in the inhibition of IL-23 production." IL 23 is certainly an important mediator in the pathogenesis of psoriasis. As we know inflammatory mediators are produced in different parts of our body due to various stimuli with some being pathogenic and others nonpathogenic. IL 23 production by dendritic cells and their role in the pathogenesis of psoriasis has not been considered the major area of IL 23's pathogenic immune dysfunction. Metformin may decrease IL23 but unless this is proven with an in vivo or in vitro model of psoriasis, one cannot make that conclusion by a "transitive" hypothesis. I think the title should be changed to " Anti-diabetic Agent Metformin Inhibits IL-23 Production 2
in Murine Bone Marrow-derived Dendritic Cells" only. If the authors wish to make a claim that this could be helpful in psoriasis as their "translational" hypothesis, they can make that statement in the discussion. Anecdotally metformin has not been very effective in diabetic patients with psoriasis and psoriasis patients are at a higher risk of metabolic syndrome. This manuscript is also more appropriate for a basic science immunology journal.